# Elucidating the Molecular Mechanisms by which Seed-Borne Endophytic Fungi, *Epichloë gansuensis*, Increases the Tolerance of *Achnatherum inebrians* to NaCl Stress

**DOI:** 10.3390/ijms222413191

**Published:** 2021-12-07

**Authors:** Chen Cheng, Jianfeng Wang, Wenpeng Hou, Kamran Malik, Chengzhou Zhao, Xueli Niu, Yinglong Liu, Rong Huang, Chunjie Li, Zhibiao Nan

**Affiliations:** 1State Key Laboratory of Grassland Agro-Ecosystems, Center for Grassland Microbiome, Lanzhou University, Lanzhou 730000, China; chengch20@lzu.edu.cn (C.C.); houwp19@lzu.edu.cn (W.H.); liuyl2020@lzu.edu.cn (Y.L.); huangr21@lzu.edu.cn (R.H.); chunjie@lzu.edu.cn (C.L.); zhibiao@lzu.edu.cn (Z.N.); 2College of Pastoral Agriculture Science and Technology, Lanzhou University, Lanzhou 730000, China; malik@lzu.edu.cn; 3State Key Laboratory of Plateau Ecology and Agriculture, Qinghai University, Xining 810016, China; 2016980005@qhu.edu.cn; 4Collaborative Innovation Center for Western Ecological Safety, Lanzhou University, Lanzhou 730000, China; 5Tibetan Medicine Research Center, College of Tibetan Medicine, Qinghai University, Xining 810016, China; 6School of Life Science and Technology, Lingnan Normal University, Zhanjiang 524048, China; niuxl05@163.com

**Keywords:** *Epichloë gansuensis*, symbiosis, full-length transcriptome, differential expression, NaCl tolerance

## Abstract

Seed-borne endophyte *Epichloë gansuensis* enhance NaCl tolerance in *Achnatherum inebrians* and increase its biomass. However, the molecular mechanism by which *E. gansuensis* increases the tolerance of host grasses to NaCl stress is unclear. Hence, we firstly explored the full-length transcriptome information of *A. inebrians* by PacBio RS II. In this work, we obtained 738,588 full-length non-chimeric reads, 36,105 transcript sequences and 27,202 complete CDSs from *A. inebrians*. We identified 3558 transcription factors (TFs), 15,945 simple sequence repeats and 963 long non-coding RNAs of *A. inebrians*. The present results show that 2464 and 1817 genes were differentially expressed by *E. gansuensis* in the leaves of E+ and E− plants at 0 mM and 200 mM NaCl concentrations, respectively. In addition, NaCl stress significantly regulated 4919 DEGs and 502 DEGs in the leaves of E+ and E− plants, respectively. Transcripts associated with photosynthesis, plant hormone signal transduction, amino acids metabolism, flavonoid biosynthetic process and WRKY TFs were differentially expressed by *E. gansuensis*; importantly, *E. gansuensis* up-regulated biology processes (brassinosteroid biosynthesis, oxidation–reduction, cellular calcium ion homeostasis, carotene biosynthesis, positive regulation of proteasomal ubiquitin-dependent protein catabolism and proanthocyanidin biosynthesis) of host grass under NaCl stress, which indicated an increase in the ability of host grasses’ adaptation to NaCl stress. In conclusion, our study demonstrates the molecular mechanism for *E. gansuensis* to increase the tolerance to salt stress in the host, which provides a theoretical basis for the molecular breed to create salt-tolerant forage with endophytes.

## 1. Introduction

Plants, as a sessile life form, are continuously exposed to environmental changes. However, they usually adapt to different environmental changes by regulating their physiological processes to balance the growth requirements and respond to extreme temperatures, excessive light, water availability, limiting nutrients, high salt environments and pathogen/insect attacks [1,2,3]. The flexible coordination of plant growth is essential to employ appropriate and rapid responses to stresses and changes in the environment [2]. As plants are sessile, they evolved a complex and precise system to enhance competitiveness and ensure their survival.

The native grass, *Achnatherum inebrians*, can adapt to harsh environments in high-altitude grassland, such as roadsides and gullies [4]. *A. inebrians*, together with the seed-borne endophytic fungi *Epichloë gansuensis,* forms a symbiotic relationship [5], which is an important factor for its adaptability and productivity to biotic and abiotic stresses [6,7,8]. This symbiosis provides several advantages to *A. inebrians*, including salt- and drought-stress resistance, enhanced water- and nitrogen-use efficiency and increased biomass [8,9,10]. Moreover, it also improves the enzyme activities of nitrogen metabolism [11,12], Cd tolerance, the ability to scavenge reactive oxygen species [13] and the resistance to disease and pests [14]. The symbiosis of plants with *Epichloë* endophytes may regulate the trade-off between the growth and resistance of host grass to abiotic stress. This may be the result of the interaction between *Epichloë* enhancing the tolerance of host grass to abiotic stress and promoting plant growth. Symbiotic plants often show more tillers and biomass compared to non-symbiotic plants [15]. The potential molecular mechanisms include *Epichloë* endophytes, which produce hormones that promote the growth of host grasses [16]; they induce morphological changes and regulation of physiological processes in the different cool-season grasses, such as altered leaf phloem, leaf xylem, stem xylem vessels, stem vascular bundles, root meta-xylem area and root system structure [17,18], as well as increasing photosynthesis [9,10,19] and enhancing the activity of glucose-6-phosphate dehydrogenase (G6PDH) [8]. High polyamines levels in wild barely may also be attributed to *Epichloë* infection [20]. The molecular mechanism of these changes caused by *Epichloë* have not yet been explored.

During the whole life cycle, plants deal with numerous environmental stresses; therefore, resource reallocation is essential to maintain their key functions, including resistance, growth and defense [16,21]. The accurate fine-tuning of stress responses and plant growth is essential in land plants. *Epichloë* endophytes (*Epichloë gansuensis*, *Epichloë inebrians*, *Epichloë festucae* and *Epichloë coenophiala*) form a mutualistic symbiotic correlation with their host grasses [7,22]. *Epichloë gansuensis* form a stable association with *Achnatherum inebrians*, an ideal host to study the interactions of native grasses and fungal endophytes. The endophyte systemically colonizes the intercellular spaces of blade tissue and leaf sheath. The *Epichloë* endophyte acquires nutrients and freedom from microbial competition from host plants, whereas host plants enhance abiotic stresses [7,23]. A study reported that the *Epichloë* endophyte is dependent on the nutrients in the intercellular spaces of plants [24]. This symbiotic relationship shows close developmental coordination, as the tissue proliferation of host grasses is generally synchronized with endophyte growth [25,26]. The function loss of some genes from *Epichloë* endophytes may occur, including *NOXA*, *NOXR* and *RACA*, which encode the NADPH oxidase complex, *PROA*, which encodes a transcription factor, and *MAPK*, which further encodes mitogen-activated protein kinase. The loss of these genes results in a severe phenotype of host–endophyte symbiosis, such as stunting and premature senescence [27,28,29,30]. The studies reported that the defense and hormonal signal transduction pathways have relationships of negative crosstalk; *Epichloë* endophytes promote plant growth via secreting plant hormones and suppress the defense responses in host grasses correlated with salicylic acid or jasmonic acid [16,31,32]. Furthermore, strong communication exists between *Epichloë* endophytes and host grasses, while endophytes up-regulate some genes of secondary metabolism in host grasses [33].

High-throughput sequencing is a vigorous method to estimate the transcriptional responses of native grasses to abiotic stresses, even without the available information on the genomic sequence. Previous studies demonstrated that many differentially expressed genes of host grasses were influenced by *Epichloë* endophytes [33,34,35,36,37]. In the current study, we explored the full-length transcriptome information of *A. inebrians*, evaluated the role of *E. gansuensis* infection on *A. inebrians* growth under 200 mM NaCl stress and identified the genes involved in generating a response to *E. gansuensis* by transcriptome strategy in the leaves of *A. inebrians* by Illumina sequencing. The current study aims to explore the transcriptome of host grasses under NaCl stress after infection with *E. gansuensis* and provide insights into the molecular mechanism that allows *E. gansuensis* to increase NaCl tolerance in host grasses.

## 2. Results

### 2.1. SMRT-Based RNA Sequencing

Single-molecule real-time (SMRT) sequencing was implemented with a PacBio sequencing platform to obtain the information of the full-length transcriptome of *Achnatherum inebrians*. The leaves and roots were collected from a single E− plant, thoroughly mixed and used for mRNA extraction. The libraries of full-length cDNA were constructed and subsequently sequenced with SMRT (Appendix A). We obtained 940,319 reads of inserts (total bases: 1,789,709,871), which included 6.41% of full-length chimeric reads and 78.55% of full-length non-chimeric reads (Appendix A). In addition, 738,588 full-length non-chimeric reads, 60,317 full-length chimeric reads and 68,878 consensus isoforms were obtained, with the average read length of consensus isoforms being 1832 (Appendix A).

### 2.2. Predictions of ORFs and SSRs

In total, 34,764 open reading frames (ORFs) were predicted for E− *A. inebrians* and the complete ORFs were 27,202. The length distributions showed that the number of ORFs with a length of 800~1200 was 5,926 (Appendix A). In most organisms, simple sequence repeats (SSR) are an important molecular marker and our results indicate that 15,945 SSRs for E− *A. inebrians* were detected and most of them were with mono-nucleotide repeats, di-nucleotide repeats, tri-nucleotide repeats and tetra-nucleotide repeats (Appendix A). Due to the high assembly quality of SMRT-derived sequences, the information of SSRs could be useful for genetic analysis and marker-assisted breeding of *A. inebrians* in the future.

### 2.3. Functional Annotation of Transcripts and Sorting of Transcription Factors

Overall, 94.16% of *A. inebrians* transcripts were annotated in selected databases and the success rate in a single database ranged from 41.54% to 93.83%. GO, KEGG, COG, KOG, Pfam, Swiss-Prot, eggNOG and NR were annotated as 82.05%, 41.54%, 42.07%, 57.65%, 79.78%, 68.45%, 92.44% and 93.83%, respectively (Appendix A). These results indicate that most of the genes were truly transcribed sequences and, probably, the functional genes in *A. inebrians*. In addition, we obtained gene families of 63 transcription factors (TFs) and identified 1536 putative TF members for *A. inebrians* (Appendix A). Further, MYB-related TFs, bHLH TFs, NAC TFs and bZIP TFs accounted for a large proportion (Appendix A).

### 2.4. E. Gansuensis Altered Transcriptome Reprogramming of A. Inebrians at Various NaCl Concentrations

To further elucidate the molecular mechanism of *E. gansuensis* on host plant growth under NaCl stress, the transcriptome of the leaves (28 days old) of E+ and E− *A. inebrians* was analyzed via Illumina sequencing. E+ *A. inebrians* and E− *A. inebrians* were treated with 0 mM NaCl and 200 mM NaCl concentrations. The thresholds FDR ≤ 0.01 and fold change ≥2 were used to determine DEGs (differentially expressed genes). The results show significant changes in the transcriptome in response to *E. gansuensis*, with 2464 and 1817 DEGs at 0 mM NaCl and 200 mM NaCl concentrations, respectively (Figure 1). In addition, significant changes in the transcriptome in response to NaCl stress, with 4919 DEGs for E+ leaves, were noted. However, a small transcriptome change occurred in response to NaCl stress, with 502 DEGs for E− leaves (Figure 1).

### 2.5. E. Gansuensis Regulated the Different Biological Processes of A. Inebrians to Maintain Balance between Growth and Resistance to NaCl Stress

*E. gansuensis* up-regulated 1269 genes and down-regulated 1195 genes of 0LE+ (leaves of E+ plants at 0 mM NaCl) compared to 0LE− (leaves of E− plants at 0 mM NaCl). Similarly, *E. gansuensis* induced 844 genes and repressed 973 genes of 200LE+ (leaves of E+ plants at 200 mM NaCl) compared to 200LE− (leaves of E− plants at 200 mM NaCl) (Figure 1). The current results imply that *E. gansuensis* had an immense influence on the regulation of gene expression at 0 mM NaCl compared with that at 200 mM NaCl. The GO enrichment analysis of DEGs between 0LE+ and 0LE− found that *E. gansuensis* mainly induced “sodium ion transmembrane transport”, “positive regulation of post-embryonic development”, “protein phosphorylation”, “photosystem stoichiometry adjustment” and “glutamate metabolic process” (Figure 2A). Besides, the GO enrichment analysis showed that the categories of “flavonoid glucuronidation”, “flavonoid biosynthetic process”, “oxidation–reduction process” and “anion transmembrane transport” were enriched among down-regulated DEGs by *E. gansuensis* at the 0 mM NaCl concentration (Figure 2B). These results imply that *E. gansuensis* reprogrammed many aspects of plant physiology and important biology process to increase the competitiveness ability of host plants. The GO enrichment analysis of DEGs between 200LE+ and 200LE− showed that *E. gansuensis* mainly induced “protein phosphorylation”, “oxidation–reduction process”, “brassinosteroid biosynthetic process”, “cellular calcium ion homeostasis” and “L-glutamate transport” (Figure 2C); furthermore, the analysis of GO enrichment found that the categories of “protein phosphorylation”, “fructose 6-phosphate metabolic process”, “flavonoid biosynthetic process” and “MAPK cascade” were enriched among down-regulated DEGs by *E. gansuensis* at the 200 mM NaCl concentration (Figure 2D). Our results show that *E. gansuensis* improved the host growth under a stress environment by activating cell signal transduction, maintaining the balance of ROS, Ca^2+^ signaling and brassinosteroid (BR) biosynthesis, which play an important role during the process of resistance to stress.

The results of the KEGG enrichment analysis found that the term “photosynthesis” was enriched among the up-regulated DEGs by *E. gansuensis* at both 0 mM NaCl and 200 mM NaCl concentrations and the terms “photosynthesis-antenna proteins” and “plant hormone signal transduction” were enriched among the up-regulated DEGs by *E. gansuensis* at 0 mM NaCl. In addition, “zeatin biosynthesis”, “arginine and proline metabolism” and “flavonoid biosynthesis” were enriched in DEGs that were up-regulated by *E. gansuensis* at the 200 mM NaCl concentration (Figure 2E). Similarly, the KEGG enrichment analysis found that the terms “starch and sucrose metabolism”, “lysine degradation” and “sesquiterpenoid and triterpenoid biosynthesis” were enriched among their down-regulated DEGs by *E. gansuensis* at both 0 mM NaCl and 200 mM NaCl concentrations and the terms “carbon fixation in photosynthetic organisms”, “tryptophan metabolism” and “pentose phosphate pathway” were enriched among the down-regulated DEGs by *E. gansuensis* at 0 mM NaCl; in addition, the terms “valine, leucine and isoleucine degradation” and “flavonoid biosynthesis” were enriched among the down-regulated DEGs by *E. gansuensis* at 200 mM NaCl (Figure 2F).

We also found that NaCl stress significantly up-regulated and down-regulated a large number of genes in *A. inebrian* LE+ and LE− (Figure 1). The analysis of GO enrichment of DEGs between 0LE+ and 200LE+ showed that NaCl stress mainly induced “flavonoid glucuronidation”, “oxidation–reduction process” and “flavonoid biosynthetic process” (Figure 3A); in addition, the categories of GO enrichment of “protein phosphorylation”, “amino acid transmembrane transport”, “flavonoid biosynthetic process”, “calcium ion transmembrane transport” and “glutamate metabolic process” were enriched among down-regulated DEGs in LE+ under NaCl stress (Figure 3B). The analysis of GO enrichment of DEGs between 0LE− and 200LE− showed that NaCl stress mainly induced “protein phosphorylation”, “amino acid transmembrane transport”, “oxidation–reduction process”, “anion transmembrane transport” and “amino acid transmembrane transport” (Figure 3C); in addition, the categories of GO enrichment of metabolism and transport of secondary metabolites (“long-chain fatty acid metabolic process”, “amino acid transmembrane transport” and “flavonoid biosynthetic process”), “oxidation–reduction process” and “potassium ion transmembrane transport” were enriched among these down-regulated DEGs in LE+ under NaCl stress (Figure 3D). These results imply that NaCl stress markedly affected the secondary metabolites’ biosynthesis, ROS signaling and amino acid metabolism of host plants.

The KEGG enrichment analysis found that the terms “galactose metabolism” and “riboflavin metabolism” were enriched among up-regulated DEGs in LE+ and LE− under NaCl stress and the terms “glyoxylate and dicarboxylate metabolism”, “carbon fixation in photosynthetic organisms” and “glycine, serine and threonine metabolism” were enriched among up-regulated DEGs in LE+ under NaCl stress; in addition, the terms “phenylpropanoid biosynthesis” and “nitrogen metabolism” were markedly enriched among up-regulated DEGs in LE− under NaCl stress (Figure 3E). Similarly, the KEGG enrichment analysis found that the term “endocytosis” was significantly enriched among down-regulated DEGs in LE+ and LE− under NaCl stress and the terms “circadian rhythm-plant”, “flavonoid biosynthesis”, “arginine and proline metabolism”, “plant hormone signal transduction” and “valine, leucine and isoleucine metabolism” were enriched among down-regulated DEGs LE+ under NaCl stress; in addition, the terms “protein processing in endoplasmic reticulum” and “nitrogen metabolism” were significantly enriched among the down-regulated DEGs LE− under NaCl stress (Figure 3F).

### 2.6. The Unique Role of E. Gansuensis on Transcriptome Reprogramming of A. Inebrians at Two NaCl Concentrations

The results of Venn showed that *E. gansuensis* only up-regulated 1135 DEGs between LE+ and LE− at the 0 mM NaCl concentration and *E. gansuensis* only up-regulated 710 DEGs between LE+ and LE− in response to 200 mM NaCl stress; in addition, 134 DEGs of LE+ vs. LE− were similarly up-regulated by *E. gansuensis* at 0 mM NaCl and 200 mM NaCl (Appendix A). Similarly, *E. gansuensis* only down-regulated 928 DEGs between LE+ and LE− at 0 mM NaCl and *E. gansuensis* only down-regulated 706 DEGs between LE+ and LE− in *A. inebrians* in response to 200 mM NaCl stress (Appendix A). The results of the analysis of GO enrichment of these special up- or down-regulated DEGs of LE+ vs. LE− at 0 mM NaCl and 200 mM NaCl, respectively, show that phosphorylation signaling, synthesis and transport of secondary metabolites, and ion transmembrane transport of *A. inebrians* were mainly enriched among unique up-regulated DEGs by *E. gansuensis* at the 0 mM NaCl concentration (Figure 4A). The results of the GO enrichment analysis show that “protein phosphorylation”, “flavonoid glucuronidation”, “amino acid transmembrane transport”, “fructose 6-phosphate metabolic process”, “flavonoid biosynthetic process” and “signal transduction by phosphorylation” of *A. inebrians* were mainly enriched among unique up-regulated DEGs by *E. gansuensis* at the 200 mM NaCl concentration (Figure 4C). The cell signal transduction pathway (“protein phosphorylation” and “cell surface receptor signaling pathway”), “flavonoid biosynthetic process”, “cell surface receptor signaling pathway”, “glucose import”, “anion transmembrane transport” and “cytokinesis” of *A. inebrians* were mainly enriched among shared up-regulated DEGs by *E. gansuensis* at both 0 mM NaCl and 200 mM NaCl concentrations (Figure 4B). *E. gansuensis* mainly only repressed “flavonoid glucuronidation”, “glucose import”, “long-chain fatty acid metabolic process”, “flavonoid biosynthetic process”, “anion transmembrane transport” and “oxidation–reduction process” of *A. inebrians*, which were mainly enriched among unique down-regulated DEGs by *E. gansuensis* at the 0 mM NaCl concentration (Figure 4D). “Protein phosphorylation”, “fructose 6-phosphate metabolic process”, “flavonoid glucuronidation”, “glycolytic process through fructose-6-phosphate”, “flavonoid biosynthetic process”, “regulation of protein phosphatase type 2A activity”, “potassium ion transmembrane transport” and “MAPK cascade” of *A. inebrians* were mainly enriched among unique down-regulated DEGs by *E. gansuensis* at the 200 mM NaCl concentration (Figure 4F). The cell signal transduction (“protein phosphorylation” and “cell surface receptor signaling pathway”), metabolism and transport of secondary metabolites (“flavonoid biosynthetic process”, “serine family amino acid biosynthetic process” and “cellular amino acid metabolic process”) and “hydrogen peroxide catabolic process” of *A. inebrians* were mainly enriched among shared down-regulated DEGs by *E. gansuensis* at 0 mM NaCl and 200 mM NaCl (Figure 4E).

### 2.7. The Unique Effect of NaCl Stress on Transcriptome Reprogramming of A. Inebrians in E+ Plants and E− Plants

NaCl stress only up-regulated 2015 DEGs between LE+ and LE+ and NaCl stress only up-regulated 139 DEGs between LE− and LE− (Appendix A). NaCl stress only down-regulated 4707 DEGs between LE+ and LE+ and NaCl stress only down-regulated 290 DEGs between LE− and LE− (Appendix A). The results of the analysis of GO enrichment of these special up-regulated DEGs at 0 mM NaCl vs. 200 mM NaCl in LE+ and LE−, respectively, show that NaCl stress mainly regulated the oxidation–reduction process, biosynthesis of secondary metabolites, small molecule transport and cell signal transduction, which were mainly enriched among unique up-regulated DEGs in LE+ and LE− (Figure 5A,C). For example, the “amino acid transmembrane transport”, “flavonoid biosynthetic process”, “potassium ion transmembrane transport”, “glucose import”, “l-α-amino acid transmembrane transport”, “anion transmembrane transport”, “signal transduction by phosphorylation”, “MAPK cascade”, “oxidation–reduction process” and “hydrogen peroxide catabolic process” of *A. inebrians* were mainly enriched among unique up-regulated DEGs in LE− under NaCl stress (Figure 5C). The metabolism and transport of secondary metabolites (“monocarboxylic acid metabolic process”, “protein catabolic process” and “lipid metabolic process”), ROS signal (“hydrogen peroxide catabolic process” and “cellular response to hydrogen peroxide”) and “protein phosphorylation” of *A. inebrians* were mainly enriched among shared up-regulated DEGs under NaCl stress between LE+ and LE−, respectively (Figure 5B). Similarly, “protein phosphorylation”, “flavonoid glucuronidation”, “flavonoid biosynthetic process”, “amino acid transmembrane transport”, “potassium ion transmembrane transport”, “calcium ion transmembrane transport” and “glutamate metabolic process” of *A. inebrians* were mainly enriched among unique down-regulated DEGs in LE+ under NaCl stress (Figure 5D). The synthesis and transport of secondary metabolites (“flavonoid glucuronidation”, “amino acid transmembrane transport”, “flavonoid biosynthetic process” and “long-chain fatty acid metabolic process”), “potassium ion transmembrane transport”, “oxidation–reduction process” and signal transduction pathways (“signal transduction by phosphorylation” and “MAPK cascade”) of *A. inebrians* were mainly enriched among unique down-regulated DEGs in LE− under NaCl stress (Figure 5F). The metabolism and transport of secondary metabolites (“amino acid transmembrane transport”, “lipid metabolic process”, “amino acid import”, “glycolytic process” and “monocarboxylic acid metabolic process”), “anion transmembrane transport”, “oxidation–reduction process” and “anion homeostasis” of *A. inebrians* were mainly enriched among shared down-regulated DEGs under NaCl stress between LE+ and LE− (Figure 5E).

### 2.8. The Effect of E. Gansuensis on the Expression Pattern of Transcription Factors of A. Inebrians under NaCl Stress

*E. gansuensis* regulated 16 transcription factors (TFs) of *A. inebrians* at the 0 mM NaCl concentration, with 10 up-regulated TFs and 10 down-regulated TFs; up-regulated TFs included MYB-related, bHLH, CAMTA, B3-ARF, C2H2, GARP-G2-like, MADS-M-type and 1 Trihelix TFs (Appendix A), with the MYB-related TFs being the most sensitive to endophytic fungi compared to other TFs at 0 mM NaCl. Down-regulated TFs included GRAS, CPP, HSF, AP2/ERF-ERF, C2C2-LSD, NAC, C3H, bHLH, bZIP and MYB-related TFs (Appendix A). *E. gansuensis* influenced 14 TFs of *A. inebrians* at the 200 mM NaCl concentration, with 5 up-regulated TFs and 12 down-regulated TFs; the up-regulated TFs, including HB-HD-ZIP, MYB, NAC, AP2/ERF-ERF, bHLH, bZIP, C2H2, C3H, GARP-G2-like, GRAS, HSF, WRKY TFs and NAC TFs, were significantly influenced by the endophyte under 200 mM NaCl stress (Appendix A); down-regulated TFs included three B3-ARF, one HB-BELL, one HB-HD-ZIP, two MYB and one NAC TFs (Appendix A). In addition, NaCl stress regulated 30 TFs of *A. inebrians* in LE+, with 23 up-regulated TFs and 17 down-regulated TFs; up-regulated TFs included AP2/ERF-AP2, C2C2-CO-like, C2C2-Dof, C2C2-LSD, CPP, FAR1, GARP-G2-like, GRAS, HB-BELL, NF-YC, RWP-RK, WRKY, zf-HD, AP2/ERF-ERF, bHLH, bZIP, C2C2-GATA, C2H2, C3H, HSF, LOB, MYB-related and NAC TFs (Appendix A). Down-regulated TFs included AP2/ERF-ERF, bHLH, bZIP, C2C2-GATA, C2H2, C3H, 2 HSF, LOB, MYB-related, NAC, B3, B3-ARF, BES1, HB-HD-ZIP, MYB, Tify and Trihelix, with MYB-related, NAC and WRYK TFs being significantly influenced by NaCl stress in LE+ (Appendix A). NaCl stress regulated seven TFs of *A. inebrians* in LE+, with two up-regulated TFs and five down-regulated TFs; up-regulated TFs included two WRKY and one NF-YC TFs, while down-regulated TFs included bZIP, GARP-G2-like, HSF, three MYB-related and one NF-YA TFs (Appendix A).

### 2.9. E. Gansuensis Alters Expression of Key Genes Associated with Flavonoid Biosynthesis, Photosynthesis, Plant Hormone Signal Transduction and Amino Acids Metabolism under NaCl Stress

*E. gansuensis* significantly activated 10 genes and 5 genes of photosynthesis at 0 mM and 200 mM NaCl concentrations, respectively. The role of *E. gansuensis* for the photosynthesis genes of the host plants was more obvious at 0 mM NaCl than 200 mM NaCl (Figure 6A). In addition, *E. gansuensis* significantly activated five genes of plant hormone signal transduction at the 0 mM NaCl concentration; these genes included two genes of the ethylene signal pathway, two genes of the auxin signal pathway and one gene of the abscisic acid signal pathway. However, *E. gansuensis* did not significantly activate genes of plant hormone signal transduction at the 200 mM NaCl concentration (Figure 6B), which implied that the effect of NaCl stress was larger than the effect of *E. gansuensis* in terms of gene expression of plant hormone. *E. gansuensis* significantly activated three genes of flavonoid biosynthesis at the 0 mM NaCl concentration, but *E. gansuensis* significantly regulated two genes of flavonoid biosynthesis at the 200 mM NaCl concentration (Figure 6C). *E. gansuensis* significantly repressed eight genes of amino acids metabolism at the 0 mM NaCl concentration and *E. gansuensis* significantly activated five genes of amino acids metabolism at the 200 mM NaCl concentration (Figure 6D). Further, *E. gansuensis* significantly down-regulated glycine, serine and threonine metabolism, as well as tryptophan metabolism, at 0 mM NaCl. *E. gansuensis* significantly down-regulated lysine degradation, up-regulated arginine and proline metabolism at 200 mM NaCl. Therefore, probably, *E. gansuensis* improves the salt tolerance of *A. inebrians* by regulating the amino acid metabolism of host grass.

Similarly, NaCl stress significantly down-regulated seven genes of photosynthesis-antenna protein in LE+ (Figure 7A). The three genes of plant hormone signal transduction involved in the ethylene, auxin and jasmonic acid signal pathways were clearly down-regulated in LE+ under NaCl stress (Figure 7B). The five genes and four genes of amino acids metabolism were up-and down-regulated in LE+ under NaCl stress, respectively (Figure 7C). Further, NaCl stress significantly down-regulated tryptophan metabolism in LE−. NaCl stress activated glycine, serine and threonine metabolism and inhibited arginine and proline metabolism in LE+. A total of eight genes of flavonoid biosynthesis were down-regulated in LE− under NaCl stress (Figure 7D).

## 3. Discussion

Our study aims to explore the full-length transcriptome information of *A. inebrians*, determine key signal pathways and identify the differentially expressed genes in the host grasses due to the presence of *E. gansuensis* symbiont at two salt concentrations. The current study provides an understanding of the molecular mechanisms and reveals that *E. gansuensis* regulates the trade-off between *A. inebrians*’ growth and resistance to salt stress. So far, only the genome information of *A. inebrians* is available, while second- and third-generation sequencing technologies have become the preferred method to explore genomic resources of non-model plants. Previous studies showed that the genomic information of tall fescue and *Lolium* species were assemblies by de novo technology [38,39]. Our study, for the first time, provides the full-length transcriptome information of *A. inebrians* by SMRT sequencing with a PacBio sequencing platform. Additionally, the transcriptome of E+ and E− *A. inebrians* at 0 mM and 200 mM NaCl were sequenced with the Illumina NovaSeq platform, to determine the role of *E. gansuensis* on host grass growth and NaCl resistance.

*E. gansuensis* can promote the growth of *A. inebrians* under abiotic stresses [7,8,10,11,12,40,41]. Despite intensive studies on the physiological mechanism between *A. inebrians*-*E. gansuensis* symbiosis and abiotic stresses, no reports have been presented to elucidate the molecular mechanism of *E. gansuensis* to increase the host grasses’ tolerance to NaCl stress. The effect of *E. gansuensis* on the trade-off of plant growth resistance through regulating DEGs and key signal pathways needs to be explored. In the present study, we found that the genes of photosynthesis were up-regulated by *E. gansuensis* at 0 mM and 200 mM NaCl. Similarly, a previous study reported that AMF increased salt tolerance of asparagus leaves, which is probably attributed to an increase in photosynthesis of asparagus [42]. However, the numbers of up-regulated DEGs of photosynthesis were greater under NaCl stress than at a 0 mM NaCl concentration, which implies that the effect of NaCl stress on the genes of photosynthesis was greater than the effect of *E. gansuensis*. The finding is consistent with the results by Alikhani et al. [43], who claimed that high photosynthesis activity resulted in a higher dry weight of E+ plants than E− plants.

*E. gansuensis* changed some gene expression of *A. inebrians* at 0 mM NaCl and up- and down-regulated some genes of host grass resistant to NaCl stress. *E. gansuensis* played an important role in the balance between host grass growth and resistance, while the endophyte in symbiosis with the host was involved in altering the gene expression strategy to adapt to salt stress. Another study demonstrated that *Epichloë coenophiala* remarkably regulated the expression of a low number of genes, which is attributed to the symbiosis with *Lolium arundinaceum* at normal growth conditions and *Epichloë coenophiala*, mainly influencing three biology processes, including response to chitin, respiratory burst during defense response and intracellular signal transduction [24]. However, in our study, *E. gansuensis* regulated cell signal transduction, positive regulation of post-embryonic development and photosystem of host grass at 0 mM NaCl. Importantly, one study surprisingly found that *Epichloë* endophyte significantly regulated over 38% of genes of the host grasses [37]. On the other hand, few studies discovered other plant–fungal relationships in which only 1–3% of the genes of the host plants were regulated by colonized fungi, such as *Arabidopsis* interacting with *Trichoderma* [44] and *Funneliformis mosseae* with tomato [45]. A study showed that *Epichloë festucae* reprogrammed the metabolism in ryegrass, especially the secondary metabolism; *Epichloë festucae* also influenced cell-wall biogenesis and trichome formation [37]. Another study found that the synthesis of secondary metabolites (fatty acid biosynthesis pathways, cutin, suberin and wax biosynthesis, carotenoid biosynthesis) were significantly enriched among the DEGs of *P. indica* at 0 mM NaCl [46]. Similarly, *E. gansuensis* regulated the cutin biosynthetic process, wax biosynthetic process and long-chain fatty acid metabolic process of the host plant based on DEGs at 0 mM NaCl. In addition, it also regulated the other biosynthesis and transport of secondary metabolites, cell signal transduction and oxidation–reduction process. In the long-term evolution process, *E. gansuensis* and *A. inebrians* coexist mutually; the effect of *E. gansuensis* on the genes expression of the host is far greater than endophyte inoculation on plant gene expression. This symbiotic relationship has a more profound impact on the host′s gene regulation. In a normal environment, few DEGs attributed to *Epichloë*’s status were consistent with ultrastructural studies of the symbioses of grass–*Epichloë*, which showed little or no alteration of plant cell architecture in the presence of *Epichloë* [26,47]. Our results are consistent with those of other studies, finding that there are large numbers of DEGs between E− plants and E+ plants [33,36,37]. The differences might be caused by different hosts, *Epichloë* endophyte, or sequencing techniques [24,33,35,36,37].

The GO enrichment analysis of DEGs showed that the flavonoid biosynthetic process in rice was activated by *P. indica* under salt stress [48]. Some early steps of the flavonoid biosynthetic process were down-regulated by the colonization with *P. indica*, while cytochrome P450 enzymes were up-regulated. This enzyme is involved in flavonoid compound production [49]; flavonoids are a class of important secondary compounds and play a key role in protecting plants from abiotic stress and promoting rhizobia colonization [50]. Our results demonstrate that *E. gansuensis* up-regulated the DEGs of flavonoids biosynthesis at the 0 mM NaCl concentration, but down-regulated them at the 200 mM NaCl concentration, which implies that the role of NaCl stress for flavonoid biosynthesis was greater than that of *E. gansuensis*. Flavonoids are an antioxidative agent and they scavenge excess ROS in plants, which are generated under abiotic stresses. Further, the ROS levels are inhibited by flavonoids mainly through inhibiting the activity of ROS-related enzymes [51]. *Enterobacter* sp. EJ01 increased ascorbate peroxidase activities in tomato under salt stress compared to EJ01-free seedlings [48]. In addition, *P. indica* also activated the biosynthesis of secondary metabolites (diterpenoid biosynthesis and carotenoid biosynthesis) under salt stress [46]. Similarly, we found that the xyloglucan biosynthetic process and carotene catabolic process were up-regulated among DEGs by *E. gansuensis* under salt stress. The above results illustrate that the biosynthesis of secondary metabolites is a very important strategy to improve the ability of the plant to resist abiotic stress.

Plant hormones are important endogenous factors, regulating plant physiological processes, to adapt to environmental change, and the plant–microorganism interaction [49,52]. For instance, among the genes of auxin biosynthesis, the *TGG2* gene was up-regulated and the *UGT1* gene was down-regulated in *Arabidopsis* when *Bacillus amyloliquefaciens* FZB42 inoculated *Arabidopsis* at 0 mM NaCl [49]. Similarly, the *ILL5* gene was up-regulated and the *UGT1* gene was down-regulated in *Arabidopsis* when *Bacillus amyloliquefaciens* FZB42 inoculated *Arabidopsis* at 100 mM NaCl [49]. The auxin signaling pathway was activated by *P. indica* under salt stress [46]. A study discussed that auxin produced by *P. indica* enhanced the root branching of *Arabidopsis* [52]. Interestingly, *E. gansuensis* up-regulated three genes involved in auxin biosynthesis, three genes involved in the ethylene signal pathway and one gene involved in the abscisic acid signal pathway in *A. inebrians* at a 0 mM NaCl concentration. However, *E. gansuensis* did not significantly regulate the genes expression of the plant hormone signal pathway of *A. inebrians* under NaCl stress. Contrary to our results, the *NCED3* (encode enzymes for abscisic acid biosynthesis) gene showed decreased expression when *Bacillus amyloliquefaciens* FZB42 inoculated *Arabidopsis* at 0 and 100 mM NaCl [49]. A study illustrated that *P. indica* activated the genes involved in zeatin metabolism (cytokinin dehydrogenase) under salt stress [46]; however, *E. gansuensis* did not influence the gene expression of cytokinin metabolism, but influenced brassinosteroid homeostasis. The above results implies that the different endophytes have different influences on the reprogrammed gene expression of plant hormones in host plants under salt stress.

However, 200 mM NaCl stress significantly decreased the number of up-regulated DEGs of host grasses between E+ and E− leaves, compared to 0 mM NaCl. However, it did not significantly influence the number of down-regulated DEGs of *A. inebrians*, compared to 0 mM NaCl. The root endophytic fungus *Piriformospora indica* increased the up- and down-regulated DEGs numbers of barely leaves at 300 mM NaCl, compared to 0 mM NaCl [53], which is inconsistent with our results. The difference is probably due to the different colonization ways of endophytic fungi, because *E. gansuensis* only colonized the above-ground tissues of *A. inebrians*, but *Piriformospora indica* colonized the root tissues of plants. The effect of NaCl stress was different for the DEGs between LE+ and LE−. In other words, *E. gansuensis* reprogrammed the gene expression of host plants to adapt to NaCl stress. NaCl stress often causes the burst of ROS, which could break the redox balance and cause oxidative damage to plants [8,54]. Interestingly, *E. gansuensis* down-regulated the DEGs of GO terms related to oxidation–reduction processes at 0 mM NaCl, but up-regulated DEGs of GO terms related to oxidation–reduction processes at 200 mM NaCl. NaCl stress mainly down-regulated the expression of DETs genes, which encode antioxidant enzymes, including POD and GST, but AMF inoculation up-regulated SOD transcript of asparagus roots under NaCl stress [55]. Further, we previously reported that *E. gansuensis* increased SOD and POD activity of *A. inebrians* under NaCl stress [8]. Therefore, it enhances host grass’ tolerance to NaCl stress through increased gene expression levels and activity of antioxidant enzymes. The above results show that *E. gansuensis* might relieve the harmful effects of NaCl stress on *A. inebrians* by increasing the ability of ROS-scavenging. In addition, salt stress significantly affected the galactose metabolism and flavonoid biosynthesis of rice [46]. In our study, NaCl stress up-regulated the arabinan catabolic process, lignin catabolic process and flavonoid biosynthetic process of the host plant and some studies illustrated these processes played an important role in helping the plant to adapt to salt stress [56,57]. Therefore, the synthesis of secondary metabolites vulnerable to salt stress and the function of secondary metabolites includes the protection from the damage of reactive oxygen species and the enhancement of the resistance of the cell wall [58]. It is well known that cell signal transduction is crucial for the tolerance of plants to salt stress and protein phosphorylation and dephosphorylation are important parts of cell signal transduction. Hence, protein phosphorylation plays a decisive role in improving plant growth under salt stress [59]. Our results show that NaCl stress down-regulated the protein phosphorylation in LE+, but up-regulated the protein phosphorylation in LE−.

Salinity usually increases ethylene content in plants, which inhibits plant growth and influences plant development [60]. Our results also show that NaCl stress down-regulated the genes of the ethylene and auxin signal pathways in the leaves of E+ plants, but NaCl stress did not significantly regulate the gene expression of the auxin and ethylene signal pathways in the leaves of E− plants. It implies that *E. gansuensis* played a central role in increasing the tolerance of host grass to NaCl stress by influencing the auxin and ethylene signal pathways. Some studies found that PGPRs produced ACC deaminase that decomposed the precursor of ethylene (ACC). Thus, PGPRs might increase plants tolerance to salt stress by decreasing the levels of ethylene under salinity stress [57,61]. *Variovorax paradoxus* 5C-2 promoted the growth of the *eto1-1* mutant but did not promote *etr1-1* and *ein2-1* growth [62]. Hence, *E. gansuensis* enhanced the tolerance of host grasses to NaCl stress probably by reducing ethylene levels. Kinetin could break the dormancy of tomato and cotton and stimulated plant growth under salt stress [63,64]. However, we found that salt stress down-regulated the gene expression of Cis-zeatin O-glucosyltransferase in LE−.

Salt stress significantly influences the nitrogen metabolic process in plants [10,65]. *P. indica* improved the growth of tobacco and *Arabidopsis* via regulating the activity of N metabolic enzymes [66]. The overexpressing of GS in rice increases its tolerance to NaCl stress [67]; meanwhile, GS also participates in proline metabolism [53]. Our previous study indicated that *E. gansuensis* enhanced the GS activity of *A. inebrians* under NaCl stress [10], which increased the host grasses’ resistance to NaCl stress. In addition, *E. gansuensis* up-regulated the genes expression of arginine and proline metabolism under NaCl stress, which was consistent with the previous research results. However, *E. gansuensis* down-regulated the gene expression of glycine, serine and threonine metabolism, as well as tryptophan metabolism, at a 0 mM NaCl concentration. The changes in amino acid concentrations are due to the presence of the endophyte [68]. Another study illustrated that NaCl stress significantly regulated glutathione metabolism in *Arabidopsis thaliana* [49]. We found that NaCl stress significantly down-regulated the nitrogen metabolism and our previous study also illustrated that NaCl markedly inhibited the activity of enzymes involved with nitrogen metabolism [10]. We also found that NaCl stress regulated the amino acid transmembrane transport in the host plant; one study reported that valine and aspartate across the PM (plasma membrane) were crucial for improving a plant’s resistance to salt stress [69,70]. Some WRKY genes were up-regulated, such as WRKY40, WRKY6, WRKY53 and WRKY33, showing the plant’s response to salt stress [71]. Our results show that NaCl stress up-regulated the genes expression of WRKY in LE+ and *E. gansuensis* induced the genes expression of WRKY under NaCl stress, showing that endophytes increased the host tolerance to NaCl stress via regulating the expression of WRKY transcript factors. In addition, WRKY1 and WRKY2 can increase resistance in barley to powdery mildew. Possibly, the lower expression of WRKY1 and WRKY2 in E+ plants, compared to E− plants, might lead to enhanced resistance of E+ plants to fungal pathogens [24].

In regard to the trade-off of plant growth–resistance related with the *Epichloë* endophyte, the resistance of E+ plants is mainly influenced by bioactive alkaloids, which are produced by host grass-*Epichloë* endophyte symbiont [16,72]. In the present study, the GO terms were mediated by *E. gansuensis* and were clearly different between 0 and 200 mM NaCl. We found that *E. gansuensis* mainly up-regulated positive regulation of post-embryonic development; embryogenesis terminates with a dormancy phase for sporophyte growth and future development. Some studies reported that few transcription factors precisely control post-embryonic seedling establishment [73]; in a non-stress environment, *E. gansuensis* mainly affects the expression of genes related to postembryonic development, which is beneficial to the healthy growth and development of plants. Galactose metabolism was induced by *E. gansuensis* at 0 mM NaCl; galactose metabolism plays a central role during the plant lifecycle [74]. Oxalate accumulation in pasture and crop plants negatively influences the nutritional quality of feed and foods and livestock eating grass with high content of oxalate content may cause poisoning and death [75]. We found that *E. gansuensis* up-regulated oxalate catabolism, which decreased the oxalate contents, benefiting pasture quality and livestock health. The wax biosynthetic process, flavonoid biosynthetic process, long-chain fatty acid metabolic process and oxidation–reduction process were down-regulated by *E. gansuensis* at 0 mM NaCl. Long-chain fatty acids are imperative to plant survival and have various functions throughout each period of plant growth and development [76,77]. Wax and cutin, as the derivative of long-chain fatty acids, are synthesized in a specific cell [76], therefore, wax and cutin decrease with the decrease in long-chain fatty acids. One study demonstrated that the cycle of oxidation–reduction works as a central hub of energy metabolism and controls ROS levels in the plant, which play crucial roles in the process of plant growth and development [78]. Our results show that *E. gansuensis* down-regulated the process of oxidation–reduction in host grass at 0 mM NaCl. However, under NaCl stress, the roles of *E. gansuensis* on the reprogrammed biological processes of host grass were different from 0 mM NaCl. Probably, these differences were the potential molecular mechanisms to explore the trade-off of growth and resistance by *E. gansuensis*. The endophyte *E. gansuensis* up-regulated the oxidation–reduction process, brassinosteroid biosynthetic process, cellular response to redox state, cellular calcium ion homeostasis, carotene biosynthetic process, positive regulation of proteasomal ubiquitin-dependent protein catabolic process, proanthocyanidin biosynthetic process, cellulose biosynthetic process and amino acids metabolic process of host grass under NaCl stress. It is well established that NaCl stress could induce osmotic and ionic stress in the plant, which results in increased ROS levels, threatening plant growth [79]. *E. gansuensis* activated the oxidation–reduction process, leading to the balance of ROS levels in plant cells. In addition, brassinosteroid (BR) are steroid hormones and play a fundamental role in plants growth and development [80]. One study showed that enhanced BR signal activity led to an increase in the tolerance to salt stress in Arabidopsis [81]. An active BR, 24-Epibrassinolide is produced from brassinolide biosynthesis, which improves plant growth and resistance by improving plants morphology under salt stress [82]. *E. gansuensis* up-regulated the BR biosynthetic process of host grass, a good strategy to resist under salt stress. The SOS signaling pathway plays a crucial function in plant resistance to salt stress. The SOS3 can function as a primary calcium sensor to bind the increased Ca^2+^ under salt stress and activate the SOS signaling pathway [83]. *E. gansuensis* induced the calcium ions to activate the SOS signaling under salt stress. The ubiquitin-proteasome system acts as an important player in plant adaptation to salinity stress and the ability of plants’ survival to salinity stress relies on the plasticity of the proteome. This system also plays a central role in helping plants to change the proteome to efficiently respond to salinity stress [84,85,86]. In our study, *E. gansuensis* significantly activated the ubiquitin-proteasome system to change the proteome of host grass to adapt to salt stress. The proanthocyanidins are the broadly distributed compounds that are involved in multiple aspects of plant growth under abiotic stress and the high accumulation of proanthocyanidins in plant was positively correlated with the tolerance of plant to salinity stress [87]. Hence, the presence of the endophyte improved the host grass growth under salt stress by activating proanthocyanidin biosynthesis. The function of a decrease in fructose-2,6-bisphosphate levels causes the increase in sucrose content; probably, this indicated that this contributes to improving the tolerance of rice to salt stress [88]. We found that *E. gansuensis* down-regulated the fructose-6-phosphate metabolic process, which probably increased the accumulation of sucrose helping host grass to resist under salt stress. Interestingly, *E. gansuensis* also activated a photosystem stoichiometry adjustment, which is the process of producing energy at 0 mM and 200 mM NaCl. The previous study reported that photosystem stoichiometry adjustments in chloroplasts enhanced the photosynthesis quantum efficiency, which provides enough energy for the growth of plants at 0 and 200 mM NaCl [89]. In addition, protein phosphorylation was affected by *E. gansuensis* at 0 and 200 mM NaCl and protein phosphorylation was catalyzed by protein kinases, a ubiquitous and important post-translational modification method for regulating the structure, activity and function of enzymes [90,91,92,93,94]. A study indicated that protein phosphorylation signaling cascades of the plant were activated under NaCl stress [59]. The current results and discussion provide an insight into molecular mechanisms of trade-off in *E. gansuensis* between the growth and resistance of host grass to salt stress. Therefore, we propose a model to illustrate the role of *E. gansuensis* in reprogramming the biological processes of host grass relatively to the trade-off between growth and resistance to salt stress (Figure 8).

## 4. Materials and Methods

### 4.1. Plant Material and RNA Extraction

E+ and E− *Achnatherum inebrians* seedlings were prepared as described in our previously reported studies [8,10]. Briefly, the seeds of *E. gansuensis*-infected (E+ plants) and non-infected (E− plants) *A. inebrians* were sown in 16 pots (lower × upper × height: 10 cm × 18 cm × 19 cm), including 8 pots with 6 E+ seeds and 8 pots with 6 E− seeds. Each pot was filled with vermiculite (150 °C for 3 h in an oven). One week after seed germination, 150 mL of 1/2 Hoagland was irrigated to each pot every week, meanwhile, each pot was irrigated with 200 mL distilled water every week. After 6 weeks, 4 E+ seedlings pots and 4 E− seedlings pots were treated for 4 weeks with 150 mL of 1/2 Hoagland with 0 mM NaCl and 4 E+ seedlings pots and 4 E− seedlings pots were treated for 4 weeks with 150 mL of 1/2 Hoagland with 200 mM NaCl; meanwhile, 4 E+ seedlings pots and 4 E− seedlings pots were treated with 200 mL distilled water every week and 4 E+ seedlings pots and 4 E− seedlings pots were treated with 200 mL distilled water with 200 mM NaCl every week. After 4 weeks of treatment at 0 mM and 200 mM NaCl concentrations, the total RNA of E− plants (leaf and root were mixed) was extracted, according to the manufacturer′s protocol, by the plant RNA extraction kit (TianGen, Beijing, China) and digested with DNase I to remove genomic DNA, which was sequenced by PacBio to explore the full-length transcriptome. Further, a total of 12 RNA samples from 3 leaves of E+ and E− plants (each treatment with 3 independent replicates) at 0 mM NaCl and 200 mM NaCl was extracted to explore the transcriptome by Illumina sequencing (Illumina, San Diego, CA, USA). The extracted RNA was checked and quantified with an Agilent bioanalyzer 2100 (Agilent, Santa Clara, CA, USA) and a Nanodrop 2000 (Thermo Fisher Scientific, Waltham, MA, USA).

### 4.2. PacBio Library Preparation, SMRT Sequencing and Raw Data Analysis

The first strand of cDNA was synthesized by the SMARTer PCR cDNA synthesis kit (Takara, Kusatsu, Japan). The sequencing of the full-length transcriptome was performed at Biomarker Technologies Co. Ltd. Briefly, the high-quality and full-length cDNAs were synthesized from RNA with a SMARTer™ PCR cDNA Synthesis Kit (Takara, Japan), after PCR amplification, the cDNA libraries were constructed with a BluePippin™ Size Selection System (Pacbio, Menlo Park, CA, USA). Then, the libraries were sequenced with PacBio RS II (Pacific Biosciences, Menlo Park, CA, USA); next, the data of Illumina sequencing were used for PacBio assembly.

The ISO-seq module in SMRTlink software v8.0 (Pacbio, USA) is used to cluster similar sequences in the full-length non-chimeric sequences (FLNC) into a cluster. Each cluster obtains a consensus isoform. To further correct the consistent sequences in each cluster, in order to obtain high-quality FL transcripts of Iso-Seq, the redundancy data were removed with cd-hit software. In order to error-correct the reads of inserts (ROIs) in PacBio data, raw data were analyzed with Iso-seq pipeline (Pacbio, USA) (with minFullPass = 3 and minPredictedAccuracy = 0.9). Next, full-length (FL), non-chemiric (NC) transcripts were identified by seeking for 5′, 3′ cDNA primers and the signal of polyA tail in ROIs. The consensus isoforms were obtained via ICE (iterative clustering for error correction) and FL consensus sequences were polished with Quiver.

### 4.3. Illumina Sequencing and Gene Annotation

The construction and sequencing of the Illumina library and its read analysis was carried out at Biomarker Technologies Co. Ltd. (Beijing, China), according to the method by Fang et al. [95], Haas et al. [96] and Yin et al. [97]. Raw reads in fastq format were performed with in-house Perl scripts. The clean data were obtained by removing reads containing ploy-N, as well as adapter and low-quality reads from raw data, with cut adapter software. The gene function was annotated based on the following databases: Nt (NCBI, non-redundant nucleotide sequences), Pfam (Protein family), KOG/COG (Clusters of Orthologous Groups of proteins), Nr (NCBI non-redundant protein sequences), Swiss-Prot (A manually annotated and reviewed protein sequence database), GO (Gene Ontology) and KO (KEGG Ortholog database).

### 4.4. Identification of Differentially Expressed Genes

In total, four groups, 0LE+ vs. 0LE−, 200LE+ vs. 200LE−, 200LE+ vs. 0LE+ and 200LE− vs. 0LE−, were compared for identification of differentially expressed genes. The DESeq software v.1.5.6 was used for the analysis of differentially expressed genes [98]. The false discovery rate (FDR) < 0.01 and fold change ≥2 were used as the screening standards.

### 4.5. The Analysis of GO Enrichment and KEGG Pathway

The enrichment analysis of Gene Ontology (GO) for DEGs was performed with the GOseq R packages, based on Wallenius non-central hyper-geometric distribution [99]. The statistical enrichment of DEGs for the KEGG pathway was tested with KOBAS software It is a standalone command-line program written in Python (2.3.4) [100]. The map of GO and KEGG terms was carried out using the OmicStudio tools at https://www.omicstudio.cn/tool, (accessed on 14 September 2021).

## 5. Conclusions

The present study provides insights into a molecular mechanism of *E. gansuensis* to increase the tolerance of host grasses to NaCl stress by exploring the full-length transcriptome information of *A. inebrians* and carrying out the transcriptome of leaves of E+ and E− plants at 0 mM NaCl and 200 mM NaCl concentrations. *E. gansuensis* have a significant impact on the reprogrammed biological process of host *A. inebrians* to adapt to various NaCl concentrations. However, *E. gansuensis* significantly up-regulated 1269 DEGs and down-regulated 1195 DEGs at a 0 mM NaCl concentration, while *E. gansuensis* significantly up-regulated 844 DEGs and down-regulated 973 DEGs at 200 mM NaCl stress, which implies that endophytic fungi reshape host’s gene expression to adapt to high salt environments. We experimented with the key biological process that determines the role of *E. gansuensis* under NaCl stress. Under NaCl stress, the GO term enrichment analysis of DEGs revealed that *E. gansuensis* mainly activated the gene expression of beneficial plant resistance processes (BR biosynthesis, protein phosphorylation, ROS, secondary metabolites, calcium ion homeostasis, ubiquitin-dependent protein catabolic, etc.); however, at 0 mM NaCl, *E. gansuensis* mainly regulated the gene expression of the beneficial plant growth process (post-embryonic development, oxalate catabolic, wax biosynthetic, flavonoid biosynthetic, long-chain fatty acid, etc). Therefore, *E. gansuensis* reprogrammed resistance related to the biology process of host grass under NaCl stress. We also found that *E. gansuensi* significantly affected the gene expression of different types of TFs at different NaCl concentrations. For example, MYB-related and bHLH TFs were highly expressed at 0 mM NaCl, whereas MYB-related, NAC and WRYK TFs were influenced by *E. gansuensi* and are beneficial to resist under NaCl stress. The genetic data gained by PacBio sequencing mostly facilitated *A. inebrians* gene annotation and further promoted the study of gene functions. In addition, understanding the role of *Epichloë* endophyte and molecule design breeding with endophytic fungi will help to develop a salt-tolerant forage in the future.

## Figures and Tables

**Figure 1 ijms-22-13191-f001:**
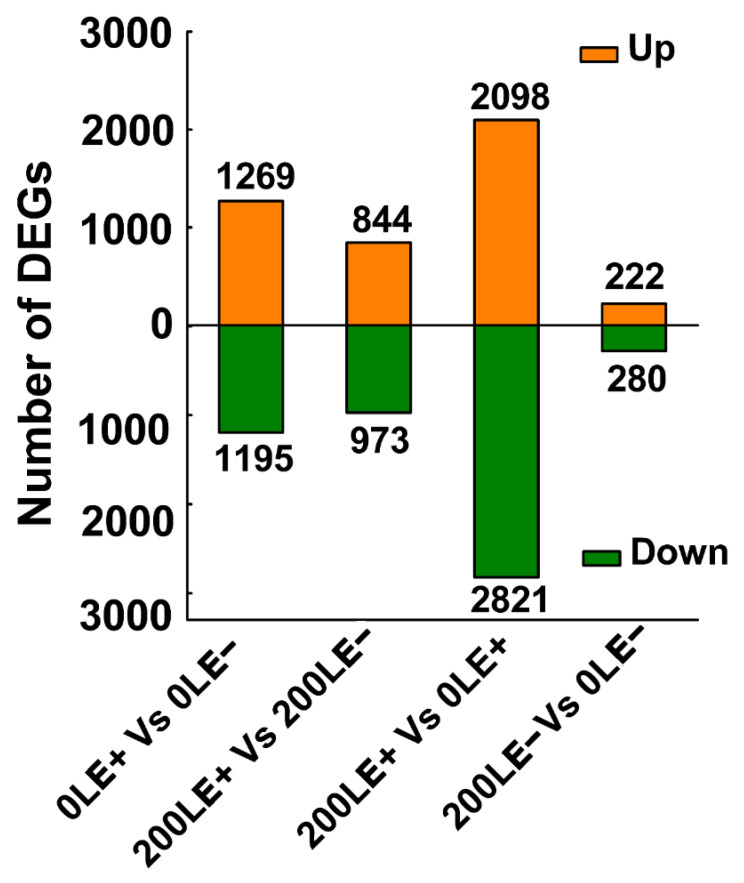
The numbers of up- and down-regulated DEGs with 0LE+ and 0LE−, 200LE+ and 200LE−, 200LE+ and 0LE+, 200LE− and 0LE−: 0LE+, leaves of E+ plants at 0 mM NaCl; 0LE−, leaves of E− plants at 0 mM NaCl; 200LE+, leaves of E+ plants at 200 mM NaCl; 200LE−, leaves of E− plants at 200 mM NaCl.

**Figure 2 ijms-22-13191-f002:**
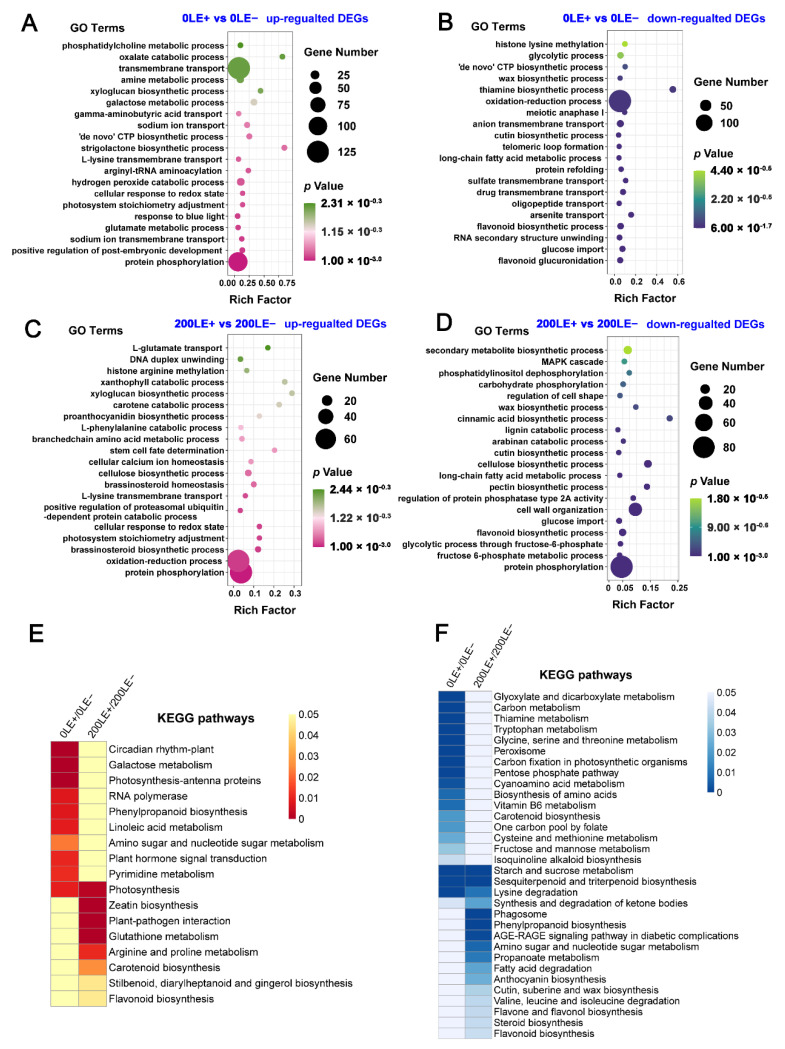
GO terms and KEGG pathway enrichment analysis of DEGs. GO terms of up-regulated (**A**) and down-regulated (**B**) DEGs identified between 0LE+ and 0LE−; GO terms of up-regulated (**C**) and down-regulated (**D**) DEGs identified between 200LE+ and 200LE−. The enrichment analysis of KEEG pathway for up-regulated (**E**) and down-regulated (**F**) DEGs between 0LE+ and 0LE− and between 200LE+ and 200LE−.

**Figure 3 ijms-22-13191-f003:**
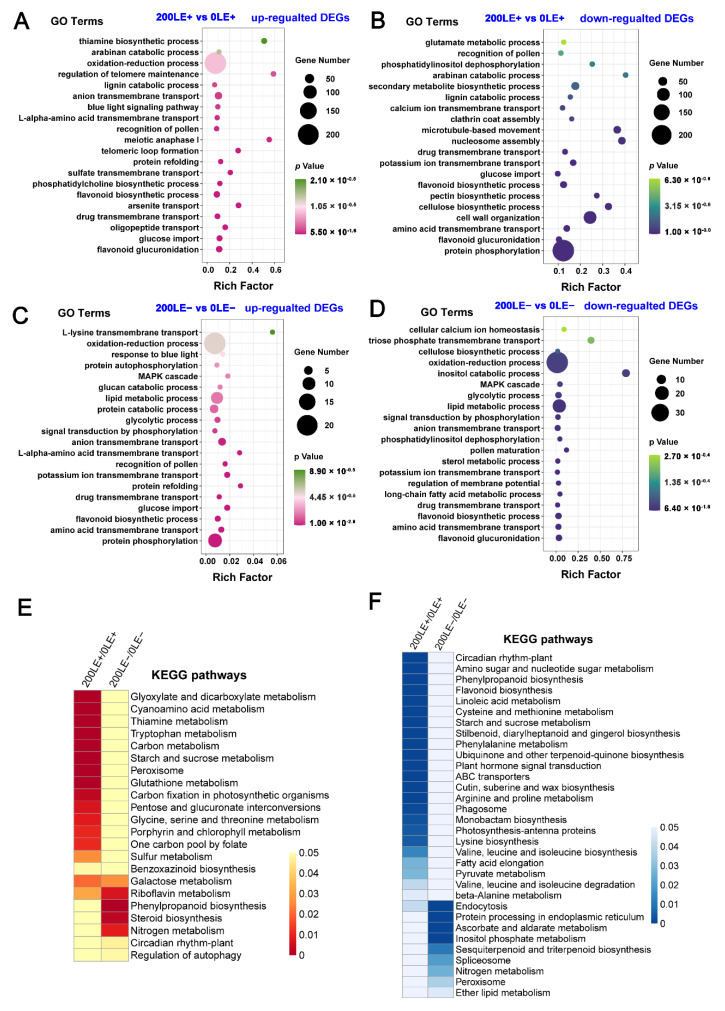
GO terms and KEGG pathway enrichment analysis of DEGs. GO terms of up-regulated (**A**) and down-regulated (**B**) DEGs identified between 200LE+ and 0LE+; GO terms of up-regulated (**C**) and down-regulated (**D**) DEGs identified between 200LE−and 0LE−. The enrichment analysis of KEEG pathway for up-regulated (**E**) and down-regulated (**F**) DEGs between 0LE+ and 0LE− and between 200LE+ and 200LE−.

**Figure 4 ijms-22-13191-f004:**
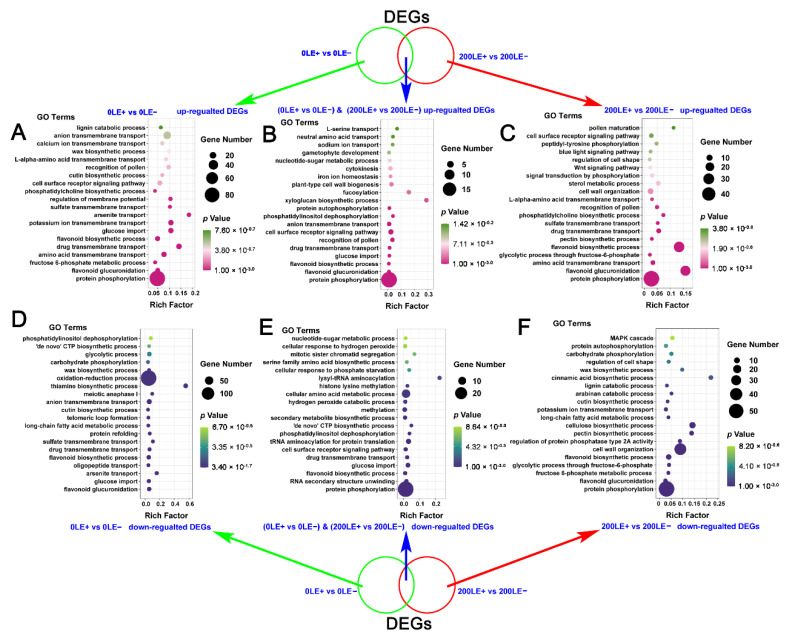
Gene Ontology (GO) categorization of non-overlapping and overlapping up- and down-regulated DEGs among the compared samples. (**A**) GO categorization of the non-overlapping up-regulated DEGs identified between 0LE+ and 0LE−; (**B**) GO categorization of the overlapping up-regulated DEGs identified between 0LE+ vs. 0LE− and 200LE+ vs. 200LE−; (**C**) GO categorization of the non-overlapping up-regulated DEGs identified between 200LE+ and 200LE−; (**D**) GO categorization of the non-overlapping down-regulated DEGs identified between 0LE+ and 0LE−; (**E**) GO categorization of the overlapping down-regulated DEGs identified between 0LE+ vs. 0LE− and 200LE+ vs. 200LE−; (**F**) GO categorization of the non-overlapping down-regulated DEGs identified between 200LE+ and 200LE−. The green circle indicates the up-regulated and down-regulated DEGs in the samples between 0LE+ vs. 0LE−; the red circle indicates the up-regulated and down-regulated DEGs in the samples between 200LE+ vs. 200LE−; the blue arrows indicate overlapping DEGs between 0LE+ vs. 0LE− and 200LE+ vs. 200LE− in the samples.

**Figure 5 ijms-22-13191-f005:**
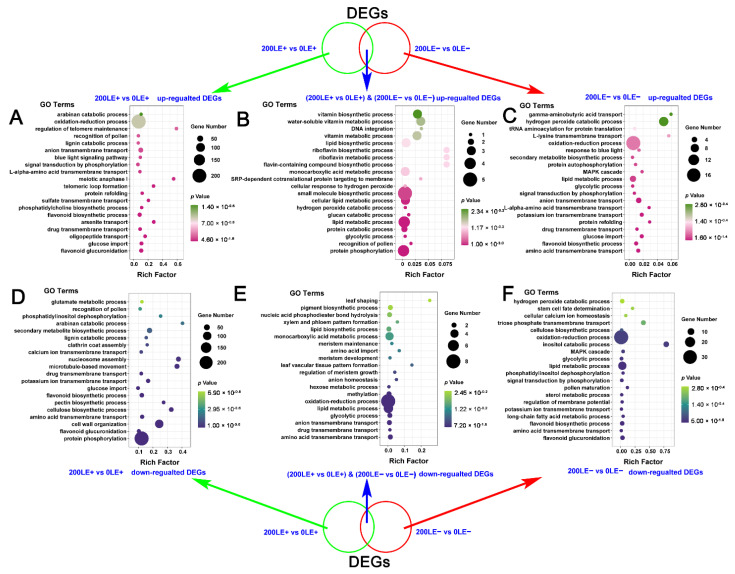
Gene Ontology (GO) categorization of non-overlapping and overlapping up- and down-regulated DEGs among the compared samples. (**A**) GO categorization of the non-overlapping up-regulated DEGs identified between 200LE+ and 0LE+; (**B**) GO categorization of the overlapping up-regulated DEGs identified between 200LE+ vs. 0LE+ and 200LE− vs. 0LE−; (**C**) GO categorization of the non-overlapping up-regulated DEGs identified between 200LE− and 0LE−; (**D**) GO categorization of the non-overlapping down-regulated DEGs identified between 200LE+ and 0LE+; (**E**) GO categorization of the overlapping down-regulated DEGs identified between 200LE+ vs. 0LE+ and 200LE− vs. 0LE−; (**F**) GO categorization of the non-overlapping down-regulated DEGs identified between 200LE− and 0LE−; The green circle indicates the up-regulated and down-regulated DEGs in the samples between 200LE+ vs. 0LE+; the red circle indicates the up-regulated and down-regulated DEGs in the samples between 200LE− vs. 0LE−; the blue arrows indicate overlapping DEGs between 200LE+ vs. 0LE+ and 200LE− vs. 0LE− in the samples.

**Figure 6 ijms-22-13191-f006:**
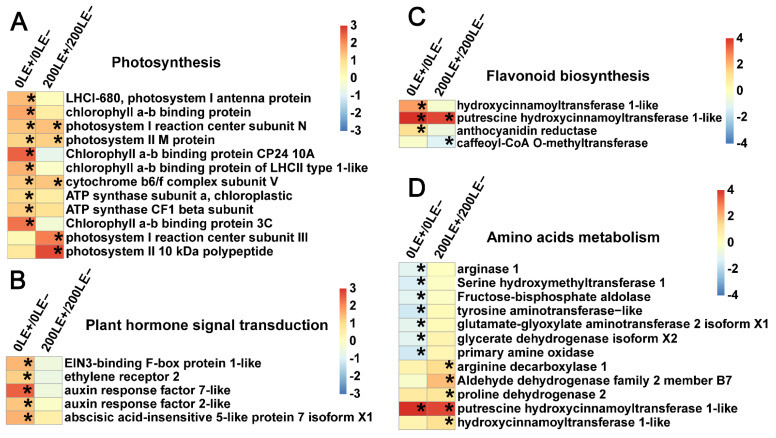
Clustering of DEGs related to photosynthesis (**A**), plant hormone signal transduction (**B**), flavonoid biosynthesis (**C**) and amino acids metabolism (**D**) between 0LE+ and 0LE− and between 200LE+ and 200LE−. Color panels mean log_2_ fold change. Asterisks represent the clear difference with FDR < 0.01 and fold change ≥2.

**Figure 7 ijms-22-13191-f007:**
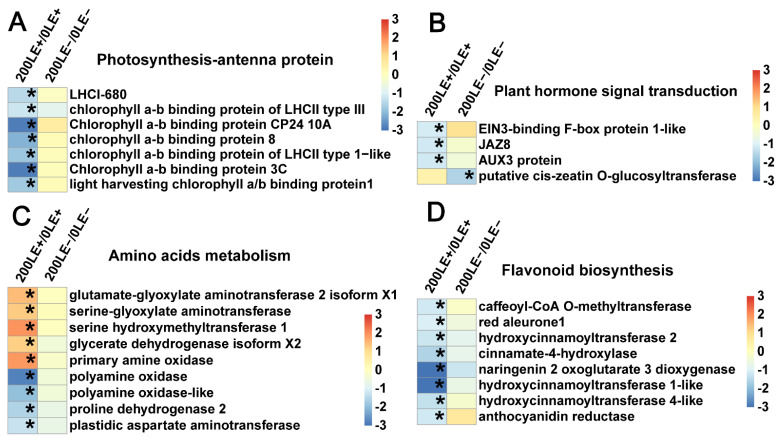
Clustering of DEGs related to photosynthesis-antenna protein (**A**), plant hormone signal transduction (**B**), amino acids metabolism (**C**) and flavonoid biosynthesis (**D**) between 200LE+ and 0LE+ and between 200LE− and 0LE−. Color panels mean log_2_ fold change. Asterisks represent the clear differences with FDR < 0.01 and fold change ≥2.

**Figure 8 ijms-22-13191-f008:**
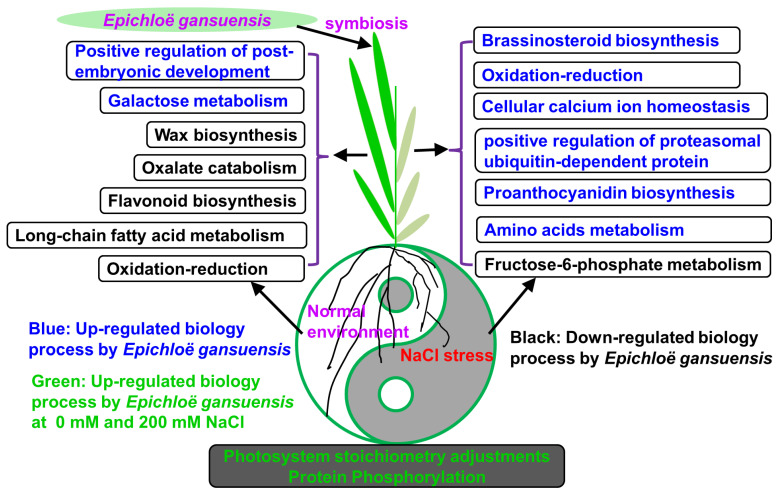
A proposed model of seed-borne endophytic fungi, *Epichloë gansuensis*, trade-off growth and salt-resistance in *A. inebrians*.

## Data Availability

Data are contained within the article.

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
