# Peer review of "Elucidating the Molecular Mechanisms by which Seed-Borne Endophytic Fungi, Epichloë gansuensis, Increases the Tolerance of Achnatherum inebrians to NaCl Stress"

_ijms, 2021, doi:10.3390/ijms222413191_

Round 1

Reviewer 1 Report

The work done on the transcriptional change of Achnatherum inebrians with Epichloë gansuensis is an important piece to understand the role of endophytes in plants under different stress conditions. The authors list many genes in the results due to the different conditions, which is summarized in the results. Many tables and graphics could be in the supplementary material to have a more structured text. It is well known, that endophytes and salt changes the gene expression and to list the numbers is redundant.

Half of the discussion is that the endophyte changes the gene expression in other studies with numbers of up and down regulated genes. The discussion would be much better, if the authors separate the effect of the endophyte on first to the host and second on the salt tolerance. It is difficult to follow the change between the two parts.  The discussion should focus on new finding to prove the trade off between stress response and plant growth.

The material and methods are not very clear how the material was prepared. After 28 days the RNA was extracted from the E- plants, what about the E+ plants? Which reads were sequenced using Pacbio and which are used for Illumina sequencing. What bioinformatic tools were used for cleaning the sequences.

Author Response

Dear Reviewer:

We have revised the manuscript “Elucidating the molecular mechanisms for seed-borne endophytic fungi, Epichloë gansuensis, increases the tolerance of Achnatherum inebrians to NaCl stress” (ijms-1421862) according to the reviewer’s comments. In our point-by-point responses attached below, reviewers’ comments are in black font and our responses are in blue font.

First of all, thank you very much for your great help and good suggestions regarding our manuscript. Through careful revision of our manuscript, some errors and grammar problems were indeed found. We are so sorry for our carelessness. But the problems now been corrected, and we hope that this manuscript is now satisfactory. In our point-by-point responses below, reviewer’s comments are in black font and our responses are in blue font

Response to Reviewer 1 Comments

Our manuscript has been critically read and revised by a native English-speaking researcher. We hope that the writing of this manuscript is now satisfactory.

Point 1: The work done on the transcriptional change of Achnatherum inebrians with Epichloë gansuensis is an important piece to understand the role of endophytes in plants under different stress conditions. The authors list many genes in the results due to the different conditions, which is summarized in the results. Many tables and graphics could be in the supplementary material to have a more structured text. It is well known, that endophytes and salt changes the gene expression and to list the numbers are redundant.

Thank you for your great help, valuable suggestions for our manuscript. The manuscript has been carefully considered, many tables and graphics were put in the supplementary material. Meanwhile, in the results and discussion, we deleted the numbers of endophyte and salt changes the gene expression in the results so that readers can understand the content more clearly.

Point 2: Half of the discussion is that the endophyte changes the gene expression in other studies with numbers of up and down regulated genes. The discussion would be much better, if the authors separate the effect of the endophyte on first to the host and second on the salt tolerance. It is difficult to follow the change between the two parts. The discussion should focus on new finding to prove the trade-off between stress response and plant growth.

Thank you for your valuable suggestion for my manuscript. In the discussion, we had separately discussed the impact of endophytes to the host plants, and second on the salt tolerance. In addition, we added the discussion with the new finding to prove the trade-off between growth and stress resistance.

Point 3: The material and methods are not very clear how the material was prepared. After 28 days the RNA was extracted from the E- plants, what about the E+ plants? Which reads were sequenced using Pacbio and which are used for Illumina sequencing. What bioinformatic tools were used for cleaning the sequences.

Thank you for your good suggestion. We are so sorry for our lack of detailed description. But the problems now been corrected, and we have provided the detail “The material and methods”. We hope that the writing of this manuscript is now satisfactory.

In fact, Epichloë gansuensis is a seed borne fungal endophyte. In this study, Neotyphodium gansuensis was isolated and identified from Achnatherum inebrians by Prof Chunjie Li from our team (Li et al 2004). In 2015, Neotyphodium gansuensis is named Epichloë gansuensis by Chen et al. (2015). Therefore, E. gansuensis was isolated and identified from A. inebrians by our team. So, E. gansuensis endophyte-infected A. inebrians (E+) and E. gansuensis endophyte-uninfected A. inebrians (E-) is used as our plant material.

In addition, the previous study showed that a survey of 20 populations of A. inebrians from low rainfall regions found that 19 populations were 100% infected with E. gansuensis (Nan and Li 2000). Therefore, E- seeds were obtained by treating with fungicide the E+ seeds. In this study, the E+ and E- seeds originated from a single E+ plants. One half seeds of a single E+ plants were as E+ seed, which were used in this study. And the other half seeds of a single E+ plants were treated with 100 times dilution thiophanate methyl for 2 h, after 2 h, the treated seeds were washed with sterile water, and they were sowed in the field, this method was the most effective way to kill the seed borne fungal endophyte of A. inebrians (Li et al. 2016). Further, we used the aniline blue staining method to check the infection status of each seedling by microscopic examination of leaf sheath pieces and seeds, uninfected seeds as E- seeds, which were used in this study. The E+ and E- seed used in this trial were as used in previous studies carried out by this research team (Li et al. 2016; Hou et al. 2020; Wang et al. 2018). Therefore, E+ and E- plants had same genetic background in the present study. This detailed information was provided by our previous study (8, 10, 101) in “Material and Methods”.

In addition, in order to avoid the influence of E. gansuensis on full-length transcriptome of A. inebrians, we used the E- A. inebrians to explore information of full-length transcriptome. Further, the leave and roots of E- plants at 0 mM NaCl were mixed, and we employed SMRT sequencing with a PacBio sequencing platform. Importantly, the transcriptome of leaves of E+ and E- A. inebrians at 0 mM and 200 mM NaCl were sequenced with the Illumina NovaSeq platform.

The cleaning sequence of SMRT raw data was used with SMRTlink and cd-hit software, and the cleaning sequence of the Illumina raw reads was used with the cut adapter software. Now, we added detailed information in the “Material and Methods”.

Chen L, Li X, Swoboda GA, Young CA, Sugawara K, Leuchtmann A, Schardl CL (2015) Two distinct Epichloë species symbiotic with Achnatherum inebrians, drunken horse grass. Mycologia 107(4):863-873.

Hou W, Xia C, Christensen MJ, Wang J, Li X, Chen T, Nan Z (2020) Effect of Epichloë gansuensis endophyte on rhizosphere bacterial communities and nutrient concentrations and ratios in the perennial grass species Achnatherum inebrians during three growth seasons. Crop Pasture Science 71: 1050-1066.

Li C, Nan Z, Paul VH, Dapprich PD, Liu Y (2004) A new Neotyphodium species symbiotic with drunken horse grass (Achnatherum inebrians) in China. Mycotaxon 90: 141-147.

Li NN, Zhao YF, Xia C, Zhong R, Zhang XX (2016) Effects of thiophanate methyl on seed borne Epichloë fungal endophyte of Achnatherum inebrians. Pratacultural Science, 33, 1306-1314.

Nan Z, Li C (2000) Neotyphodium in native grasses in China and observations on endophyte/host interactions. Proceedings of the 4th international neotyphodium-grass interactions symposium, Soest.

Wang JF, Nan Zb, Christensen MJ, Li CJ (2018) Glucose-6-phosphate dehydrogenase plays a vital role in Achnatherum inebrians plants host to Epichloë gansuensis by improving growth under nitrogen deficiency. Plant and Soil 430 (1-2):37-48.

In the end, we thank the editor and the reviewers again and hope that the revised manuscript is now acceptable.

Best regards.

Reviewer 2 Report

The review on the publication by Cheng et al. under the title Elucidating the molecular mechanisms for seed-borne endophytic fungi, Epichloë gansuensis, increases the tolerance of Achnatherum inebrians to NaCl stress

The publication contains new and interesting results demonstrating the molecular mechanisms for seed-borne endophytic fungi.

My significant points to the publication:

The results part is the part that is very difficult to follow. Authors have to summarize the data to make it easier for the reader to understand. Some of the data have to be transferred to the Supplementary material.

As the publication is very descriptive, it is not easy to follow it. I would ask the authors to prepare the scheme that demonstrates molecular mechanisms for seed-borne endophytic fungi.

I would recommend writing the conclusion into the conclusion part, not the number of overexpressed genes.

Author Response

Dear Reviewer:

We have revised the manuscript “Elucidating the molecular mechanisms for seed-borne endophytic fungi, Epichloë gansuensis, increases the tolerance of Achnatherum inebrians to NaCl stress” (ijms-1421862) according to the reviewer’s comments. In our point-by-point responses attached below, reviewers’ comments are in black font and our responses are in blue font.

First of all, thank you very much for your great help and good suggestions regarding our manuscript. Through careful revision of our manuscript, some errors and grammar problems were indeed found. We are so sorry for our carelessness. But the problems now been corrected, and we hope that this manuscript is now satisfactory. In our point-by-point responses below, reviewer’s comments are in black font and our responses are in blue font.

Response to Reviewer 2 Comments

Our manuscript has been critically read and revised by a native English-speaking researcher. We hope that the writing of this manuscript is now satisfactory.

Point 1: The results part is the part that is very difficult to follow. Authors have to summarize the data to make it easier for the reader to understand. Some of the data have to be transferred to the Supplementary material.

Thank you for your great help and good suggestions regarding our manuscript. We have revised and simplified the content of the “Results” according to your suggestions. The manuscript has been carefully considered, many tables and graphics were put in the supplementary material.

Point 2: As the publication is very descriptive, it is not easy to follow it. I would ask the authors to prepare the scheme that demonstrates molecular mechanisms for seed-borne endophytic fungi.

Thank you for your good suggestions for my manuscript. We have carefully examined our manuscript and have improved the “Material and Methods”, “Results”, “Discussion” and “Conclusion” to make the contents clearer for the readers, in addition, I provided the “A proposed model of seed-borne endophytic fungi, Epichloë gansuensisan,trade-off growth and salt-resistance in A. inebrians”, and we hope that this manuscript is now satisfactory.

Point 3: I would recommend writing the conclusion into the conclusion part, not the number of overexpressed genes.

Thank you for your valuable advises. We have revised and simplified the conclusion part in the manuscript, and deleted unnecessary descriptions of the number of genes, hope this part is more conclusive and easier to read now.

In the end, we thank the editor and the reviewers again and hope that the revised manuscript is now acceptable.

Best regards.

Round 2

Reviewer 2 Report

Dear Authors, 

I'm happy with all corrections provided by the authors.